# Effect of Dielectric Layer on Miniaturized Patch Antenna Sensor

**DOI:** 10.3390/s24237608

**Published:** 2024-11-28

**Authors:** Caifeng Chen, Lei Zou, Chenglong Bi, Andong Wang

**Affiliations:** School of Materials Science and Engineering, Jiangsu University, Zhenjiang 212013, China; chenjsust@163.com (C.C.); 15137269045@163.com (L.Z.); 13856935469@163.com (C.B.)

**Keywords:** antennas, dielectric constant, dielectric losses, dielectric layer

## Abstract

Miniature patch antenna sensors have great potential in the field of structural health monitoring for crack propagation detection due to their small size and high sensitivity. A primary research focus has been achieving efficient miniaturization, with the performance of the dielectric layer playing a pivotal role. Studies have demonstrated that increasing the relative dielectric constant (ε_r_) of the dielectric layer can reduce antenna size, but higher dielectric losses (tanδ) can lower radiation efficiency. This study identifies the optimal dielectric properties by examining the interplay between ε_r_ and tanδ to balance size reduction and radiation efficiency. Additionally, while increasing the dielectric layer’s thickness improves bandwidth and radiation efficiency, a thinner layer is preferred to maintain overall performance without compromising radiation efficiency. Furthermore, the resonant frequency of the smaller-sized patch antenna sensor exhibits greater detection sensitivity to crack propagation. These insights provide useful guidance for selecting effective dielectric layers and assist in the miniaturization design of antenna sensors.

## 1. Introduction

In industrial applications such as aircraft and industrial robots, components often endure harsh conditions that lead to strain concentration and eventual damage accumulation, initiating cracks that can result in catastrophic structural failures and significant losses [1,2]. Real-time strain measurement of critical structures is therefore crucial for detecting potential crack initiation areas promptly, ensuring the safety of essential equipment and engineering facilities. Traditional strain sensor networks typically involve extensive cabling, which adds weight, complexity, installation costs, and vulnerability to system failures during disasters due to damaged leads [3]. Antenna sensors offer a promising solution by integrating wireless capabilities with sensing functions [4,5]. Their simple installation and structure have expanded the application across various fields. Recent research has demonstrated the reliability of antenna sensors in detecting crack propagation in structural components [6,7]. For example, Mohammad et al. proposed the basic structure and operational principles of antenna sensors for crack propagation detection [8]. Sofi et al. studied the feasibility of a wireless patch antenna sensor network for structural health monitoring [9,10]. Xue et al. designed a double-layer slotted circular patch antenna to measure the maximum strain of components according to the relationship between strain and antenna resonance frequency [11]. However, challenges persist when applying antenna sensors to detect damage in complex micro-components, potentially compromising the safety and reliability assessments of precision engineering structures due to size mismatches. Thus, there is an urgent need for sensor miniaturization.

To address this, researchers have proposed many structural improvements for the miniaturization of antenna sensors. Zhou et al. proposed an effective design method for compact bandpass filters with extensible orders/channels on a modified patch resonator [12]. Santhanam et al. found that digging holes in the ground plate can also reduce the antenna resonance frequency, thus realizing the miniaturization of the sensor size [13]. Merlin et al. achieved sensor miniaturization by slotting the patch antenna surface. Although this approach reduces the antenna’s radiation surface area, it complicates antenna manufacturing and impedance matching [14]. Goswami utilized fractal antenna principles to reduce size while introducing multi-frequency characteristics, though increasing fractal complexity can compromise radiation efficiency and other antenna performance metrics [15]. Therefore, these structural improvements will reduce the radiation surface area of the antenna and affect the performance.

Recently, Singh et al. established a relationship between dielectric layer materials and antenna sensor size, showing that a higher dielectric constant of the dielectric layer results in smaller antenna sizes under required initial frequencies [16]. Miniaturized patch antenna sensors can be achieved by adjusting the dielectric layer material without modifying the antenna structure. However, dielectric materials with high dielectric constants often exhibit high dielectric losses, which can negatively impact the radiation efficiency of the antenna sensor [17,18]. Currently, the specific effects of dielectric layer properties on radiation efficiency remain underexplored. Although the implementation of miniaturized antenna sensors is feasible, it is still a challenge to ensure the performance of the dielectric electrical layer to meet the demand for crack propagation detection in small-sized antenna sensors. This work analyzes the effects of the dielectric layer on patch antenna size and radiation efficiency through simulations. This study aims to optimize dielectric layer materials to achieve antenna sensor miniaturization while enhancing operational efficiency.

## 2. Size Calculation of Antenna Sensor

The structure of the rectangular patch antenna sensor is shown in Figure 1, including the patch antenna on the upper surface, the dielectric layer in the middle, and the metal ground plane below. Through the action of the dielectric layer, an electromagnetic resonance cavity is formed between the patch antenna and the metal ground plate to receive external microwave frequency modulation signals. The electromagnetic waves form stable oscillations in the antenna and reflect the signals to obtain the return loss curve and center frequency. The center frequency of the patch antenna sensor is related to its structure, and the center frequency will also shift when the grounding plane has crack propagation. Therefore, the measured metal structure can be used as the metal ground plate, and then the strain can be characterized by the change in the center frequency of the patch antenna to detect crack propagation.

Based on the transmission model of the antenna, the length L and width W of the rectangular radiation patch were designed using Equations (1)–(4) [19].
(1)$$L=c/\left(2f\sqrt{\varepsilon_{e}}\right)-2∆L$$
(2)$$W=\left(c/2f\right){\left(\left(\varepsilon_{r}+1\right)/2\right)}^{-1/2}$$
(3)$$\varepsilon_{e}=\left(\varepsilon_{r}+1\right)/2+\left(\varepsilon_{r}-1\right)/2{\left(1+12h/w\right)}^{\left(-1/2\right)}$$
(4)$$∆L=0.412h(\varepsilon_{e}+0.3)(W⁄h+0.264)/(\varepsilon_{e}-0.258)(W⁄h+0.8)$$
where c is the speed of light in vacuum, f is the operating frequency, h is the thickness of the dielectric layer, ε_r_ is the relative dielectric constant of the dielectric layer, ε_e_ is the effective dielectric constant, and ΔL is the extension length.

Take the coordinate origin as the center of the patch antenna. The antenna length is along the *x*-axis and the width is along the *y*-axis. The central feeding method is used in this study, and it is only necessary to keep the feeding point on the *x*-axis. The distance of the feed point from the center of the patch (L_0_) will affect the impedance matching of the antenna. The position of the feeding point (L_0_, 0) can be calculated by Equation (5) [20].
(5)$$L_{0}=L/\left(2\sqrt{\varepsilon_{e}}\right)$$

Arlon CuClad233, Krempel AKaflex kcl, Rogers TMM4, Rogers TMM6, and Arlon AR1000 were used as dielectric layer materials for the patch antenna sensor. The values of the relative dielectric constant (ε_r_) and dielectric losses (tanδ) are shown in Table 1.

The center frequency is set to 3 GHz, and the thickness of the dielectric layer is 1.5 mm. The antenna dimensions are calculated using Equations (1)–(5), followed by optimization using the commercial software Ansys High Frequency Structure Simulator (HFSS version 15.0). The optimization process aims to refine the antenna dimensions. The return loss curve (S_11_), standing wave ratio curve (VSWR), and radiation efficiency are derived for the antenna.

## 3. Results and Discussion

### 3.1. Effect of Relative Dielectric Constant of Dielectric Layer on Miniaturization of Antenna Sensor

The dimensions of the coaxial patch antenna sensor can be calculated by the above five kinds of dielectric layer materials with different relative dielectric constants, and the optimized dimensions are shown in Table 2.

As can be seen from Table 2, the size of the antenna is inversely proportional to the ε_r_ value of the dielectric lay materials. When the ε_r_ value increases from 2.33 to 10, the area of the patch antenna sharply decreases from 1201.25 mm^2^ to 318.73 mm^2^. Since the coaxial antenna sensor uses a center feed, the resonant frequency of the antenna sensor is mainly affected by the antenna length L and the feed position L_0_, and it is unnecessary to change the antenna width W during optimization. Thus, compared with the calculated value, the optimized antenna length (L) is slightly reduced, while the width (W) is still consistent with the calculated value. However, Table 2 also shows that as the area of the antenna shrinks, the radiation efficiency of the antenna decreases from 98.02% to 81.57%.

From Table 2, it can also be seen that the antenna’s feeding point positions (L_0_) were determined to be 5.5, 4.2, 3.6, 2.8, and 2.0 mm, respectively. As the size of the antenna sensor decreases, the L_0_ value decreases accordingly. The location of the L_0_ mainly affects the antenna’s input impedance. Through optimization, the optimal L_0_ value is determined to achieve impedance matching with the input impedance of the antenna sensor (Z_0_ is 50 Ω), as described in reference [21].

Figure 2 shows the optimized reflection coefficient of the sensor using five kinds of materials with different relative dielectric constants of the dielectric layer, including the return loss curve (S_11_) and the standing wave ratio curve (VSWR). Figure 2a demonstrates that after optimization, the center frequency of the antenna aligns with the calculated resonant frequency of 3 GHz. The bandwidth is defined as the frequency range of the S_11_ curve within −10 dB. It can be seen from the figure that as the value of ε_r_ increases, the bandwidth gradually becomes narrower. Figure 2b displays VSWR curves of different dielectric layer materials that are relatively close, and the highest value is not more than 1.2. It means that the antenna dimensions optimized according to Table 2 satisfy the specified requirements, achieving excellent impedance matching.

It can be concluded that a high relative dielectric constant of the dielectric layer significantly impacts the size of the antenna sensor, effectively reducing the antenna’s area. However, this also results in a narrower bandwidth and decreased radiation efficiency.

### 3.2. Effect of Dielectric Loss of Dielectric Layer on Miniaturized Antenna Sensor

To further investigate the effect of dielectric losses of the dielectric layer on the size and performance of antenna sensors, we used the same relative dielectric constant to simulate and analyze the performance of antenna sensors by changing the dielectric losses of the dielectric layer. Figure 3 shows the optimized reflection coefficient of the antenna sensor after optimization. Table 3 shows the dimensions, radiation efficiency, and bandwidth of these patch antenna sensors.

Figure 3a illustrates that an increase in dielectric losses can improve the bandwidth of the antenna, and Figure 3b shows that the antenna impedance has reached a good match. Combined with Figure 3, as may be seen from Table 3, the change in dielectric loss does not affect the dimensions of the antenna sensor. However, when the relative dielectric constant of the dielectric layer is the same, the radiation efficiency of the antenna sensor decreases from 59.25% to 40.65% as the dielectric losses increase. This confirms that the dielectric layer with large dielectric losses cannot efficiently utilize the energy of the incident signal. As the dielectric losses increase, the peak gain of the antenna sensor decreases from 3.39 dB to 2.34 dB, indicating that increasing the dielectric losses is also not conducive to the long-distance application of the antenna sensor.

Therefore, we know that high ε_r_ values of dielectric layers are mainly beneficial to the miniaturization of the antenna sensor size, but the dielectric losses affect the antenna bandwidth, peak gain, and radiation efficiency. Selecting appropriate dielectric layer materials with appropriate ε_r_ and tanδ values can not only miniaturize the size of the antenna sensor but also obtain good performance.

### 3.3. Relationship Between Relative Dielectric Constant and Dielectric Losses of Dielectric Layer at Specific Radiation Efficiency

The relationship between the radiation efficiency (η), relative dielectric constant (ε_r_), and dielectric losses (tanδ) of antenna sensors can be explained by Equations (6) and (7) [22].
(6)$$\eta =\frac{P_{rad}}{P_{rad}+P_{loss}}$$
(7)$$P_{loss}=\frac{1}{2}{\left|E\right|}^{2}\frac{Whtan\delta }{240}\sqrt{\varepsilon_{r}}$$
where P_rad_ is the radiation efficiency of the antenna, P_loss_ is the loss power of the antenna, |E| is the electric field strength, W is the width of the antenna, and h is the thickness of the dielectric layer. As may be seen from the above equation, radiation efficiency (η) is inversely proportional to dielectric loss (tanδ). Specifically, an increase in dielectric loss usually leads to a decrease in radiation efficiency. Therefore, by optimizing the relative dielectric constant and loss factor of the dielectric layer, the radiation efficiency of the antenna can be improved. The following is the actual verification process.

By changing the relative dielectric constant and dielectric losses of the dielectric layer material, the specific radiation efficiency of the antenna sensor is maintained at 40%, 50%, and 60%, respectively. Table 4 shows the optimized parameters of the antenna for different layer materials.

It can be seen from Table 4 that the varying relative dielectric constant and dielectric losses of the dielectric layer will affect the length (L) of the antenna and the position of the feeding point at different specified radiation efficiencies. In contrast, the width (W) of the antenna is independent of the dielectric losses of the dielectric material. As the relative dielectric constant of the dielectric layer increases, lower dielectric losses are required to achieve the same radiation efficiency. Therefore, when determining the radiation efficiency, there may be a certain relationship between the relative dielectric constant of the dielectric layer and the required dielectric losses.

Figure 4 illustrates the S_11_ curve and VSWR curve of antenna sensors with varying specified radiation efficiency. The size optimization of the patch antenna results in impedance matching, with values consistently below 1.2 for VSWR. The S_11_ curve further indicates that as the relative dielectric constant of the substrate material increases, the bandwidth of the antenna sensor decreases. This observation aligns with the findings from simulations discussed in the preceding section.

Figure 5a–c show the data and fitting curves for the predetermined radiation efficiency of 40%, 50%, and 60%, respectively. By linear fitting, the relationship between the relative dielectric constant and critical required dielectric losses is obtained, as shown in Equations (8)–(10).
(8)$$y=1.04\times 10^{-4}\times X^{2}-3.85\times 10^{-3}\times X+0.0464$$(9)$$y=7.2\times 10^{-5}\times X^{2}-2.66\times 10^{-3}\times X+0.0321$$(10)$$y=5.6\times 10^{-5}\times X^{2}-1.97\times 10^{-3}\times X+0.0225$$

The fitting degrees for Equations (8)–(10) are 98.6%, 99.4%, and 99.1%, respectively. It indicates that the fitting equation has high accuracy. 

To further verify the accuracy of fitting equations (8)–(10) shown above, three materials with relative dielectric constants of 5, 10, and 15 were used for validation. Table 5 shows the verify parameters. 

From the simulation results presented in Table 5, the critical required dielectric losses of the antenna layer material can be calculated, and the obtained radiation efficiency closely matches the specified radiation efficiency, labeled as “R” in Table 5. Thereby, the accuracy of the fitting Equations (6)–(8) is verified. 

### 3.4. Effect of Dielectric Layer Thickness on Miniaturized Antenna Sensor

Figure 6 presents the S_11_ and VSWR curves of the antenna sensors with Rogers TMM4 dielectric layer thicknesses of 1 mm, 1.5 mm, 2 mm, 3 mm, and 5 mm, respectively. As illustrated in Figure 6a,b, the optimization process has successfully aligned the resonant frequencies with the predetermined operating frequency of 3 GHz, while the VSWR values remain below 1.2. This indicates a strong impedance match for the antenna across the various dielectric layer thicknesses, suggesting that the antenna design can achieve optimal performance. Furthermore, Figure 6a reveals that the bandwidth of the antenna increases obviously as the thickness of the dielectric layer increases.

Table 6 presents the optimized antenna dimensions for varying dielectric layer thicknesses. It can be observed that the thickness of the dielectric layer has a minimal impact on the antenna length. Specifically, as the thickness increases from 1 mm to 5 mm, the length decreases only from 23.20 mm to 21.05 mm. And the dielectric layer thickness does not influence the antenna width, which is further supported by Equation (2), demonstrating that the width remains independent of the dielectric thickness. Consequently, while the overall size of the antenna decreases slightly with increasing dielectric layer thickness, the reduction is limited. This suggests that although increasing the thickness of the dielectric layer has little effect on antenna size, it can enhance the antenna’s bandwidth obviously.

Figure 7a shows the relationship between dielectric layer thickness and antenna radiation efficiency. The data clearly indicate that as the thickness of the dielectric layer decreases, the antenna’s radiation efficiency declines significantly. When the thickness is reduced from 5 mm to 1 mm, the radiation efficiency decreases from 80% to 40%. The magnitude of radiation efficiency is closely linked to the distribution of the electric field within the dielectric layer [23]. Figure 7b illustrates the electric field distribution across dielectric layers of varying thicknesses. As may be seen from the figure, the electric field in the thick dielectric layer is more evenly and widely distributed, whereas the electric field in the thin dielectric layer is relatively concentrated in a small area. A wide electric field distribution within the dielectric layer enhances the radiation of the sensor through the edge effect of the patch antenna [24,25]. Consequently, the thick dielectric layer maintains a high radiation efficiency. However, as the dielectric layer thickness continues to increase, antenna losses rise due to the generation of surface waves [26,27], causing the growth of radiation efficiency to decelerate, as depicted by the arrow in Figure 7a.

As the thickness of the dielectric layer decreases, the electric field distribution becomes more concentrated, but the electric field intensity increases, as illustrated in Figure 7b. The high electric field intensity contributes to the generation of stable electromagnetic waves. This stability is crucial for enhancing the overall performance of the antenna, particularly in applications requiring high sensitivity. Therefore, although antenna sensors with thinner dielectric layers exhibit lower radiation efficiency, they are more capable of generating stable electromagnetic waves. For antenna sensors, it is advisable to select the thinnest possible dielectric layer while ensuring adequate radiation efficiency.

### 3.5. Effect of Crack Propagation on Miniaturized Antenna Sensor

Figure 8a–c illustrate three antenna sensors of different sizes with Arlon CuClad233, Rogers TMM4, and Arlon AR1000 dielectric layers, respectively. Their operating frequency is at 3 GHz. The relative dielectric constant of the three dielectric layers is 2.33, 4.50, and 10.00, respectively, as detailed in Table 1. The figures demonstrate a clear reduction in antenna size as the relative dielectric constant of the dielectric layer increases. Figure 8d–f present the relationship between the resonant frequencies of the three antenna sensors and crack propagation along the width direction of the metal ground plane. These figures show that as the crack expands from 0 mm to 10 mm, the resonant frequency shifts from 3 GHz towards lower frequencies.

The presence of cracks in the metal ground plane alters the current distribution within the antenna sensor, thereby impacting its electromagnetic characteristics. When a crack exists in the metal ground plane, the resonant frequency of the antenna sensor can be expressed as follows [28]:(11)$$f_{\left(crack\right)}=\frac{c}{2\sqrt{\varepsilon_{e}}}\frac{1}{L+2∆L+∆L_{\left(crack\right)}}$$

Here, c represents the speed of light in a vacuum, ε_e_ denotes the effective dielectric constant, L is the length of the antenna sensor, ΔL is the extension length, and ΔL_(crack)_ is the additional length of the current path caused by the presence of the crack.

As shown in Equation (11), crack propagation reduces the resonant frequency (f_(crack)_) of the antenna sensor, resulting in a leftward shift of the curve, as illustrated in Figure 8d–f. Notably, smaller antenna sensors exhibit greater sensitivity to changes in resonant frequency when the crack on the metal ground plane extends beyond a certain threshold. As observed in Figure 8f, the resonant frequency of the small-sized sensor decreases significantly from 3 GHz at f_(0)_ to 2.69 GHz at f_(10)_ as the crack expands from 0 mm to 10 mm. In contrast, the resonant frequency of the large-sized sensor shows minimal change with crack growth. In Figure 8d, its resonant frequency (f_(10)_) is 2.92 GHz when the crack length extends to 10 mm. This indicates that the smaller-sized antenna sensor demonstrates improved detection sensitivity.

Figure 8g–i present the surface current density profiles on the ground planes of antenna sensors with varying sizes. It can be observed that the smaller-sized sensor exhibits stronger and more uniform current density in its central region. This heightened current density leads to a more pronounced change in the electromagnetic field distribution around the ground plane. When a crack appears, the interaction between this intensified electromagnetic field and the crack generates more significant signal changes, enabling the antenna sensor to detect the presence of the crack with greater sensitivity.

## 4. Conclusions

The dielectric properties of the dielectric layer are crucial for the miniaturization and performance of antenna sensors, as demonstrated through simulations and analyses. Increasing the relative dielectric constant (ε_r_) of the dielectric layer significantly reduces the size of the antenna sensor and narrows the bandwidth. However, higher dielectric losses (tanδ) in the dielectric layer reduce radiation efficiency and antenna gain. Establishing a precise relationship between ε_r_ and tanδ enables accurate prediction of both antenna size and radiation efficiency. For radiation efficiencies of 40%, 50%, and 60%, the relationship between the relative dielectric constant and critical dielectric losses aligns with Equations (8)–(10), with validation under critical conditions confirming the accuracy of this relationship. Understanding this dynamic facilitates the selection of appropriate dielectric layer materials, thereby enhancing the performance of miniaturized patch antenna sensors. Additionally, increasing the thickness of the dielectric layer improves both bandwidth and radiation efficiency, though it has minimal impact on the antenna size. On the premise of ensuring the radiation efficiency, a thinner dielectric layer can be selected as far as possible. When crack propagation occurs in the metal ground plane, smaller-sized sensors exhibit higher sensitivity to changes in resonant frequency. These insights provide valuable guidance for designing miniaturized patch antennas using suitable dielectric materials. This advancement not only broadens the application potential of high-performance antenna sensors in structural health monitoring for crack propagation detection but also extends their use to fields such as wireless communication, radar, electronic countermeasures, and more.

## Figures and Tables

**Figure 1 sensors-24-07608-f001:**
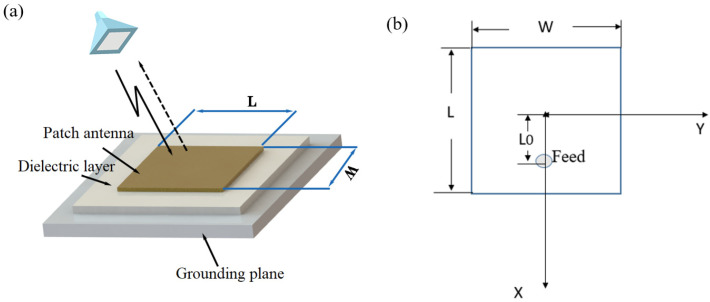
Structure of a coaxial antenna sensor: (**a**) size of the antenna; (**b**) position of the feed point.

**Figure 2 sensors-24-07608-f002:**
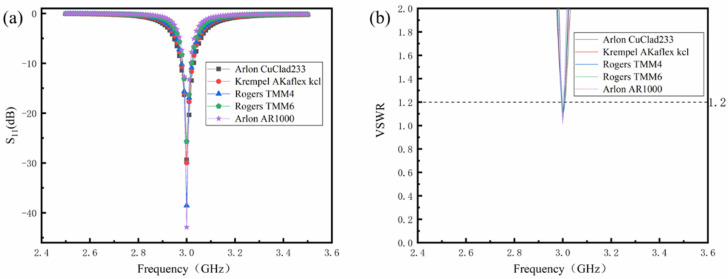
The optimized reflection coefficient of antenna sensors with varying relative dielectric constants of the dielectric layer: (**a**) S_11_; (**b**) VSWR.

**Figure 3 sensors-24-07608-f003:**
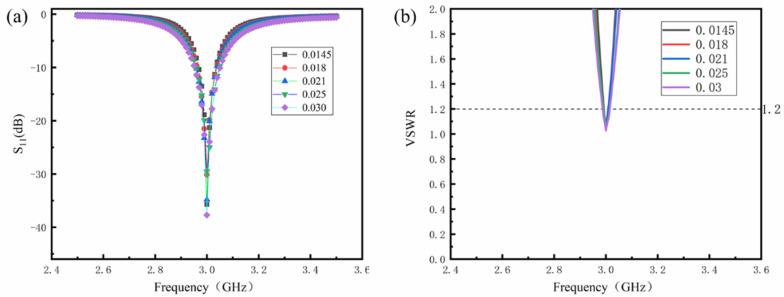
The optimized reflection coefficient of antenna sensors with varying dielectric losses of the dielectric layer: (**a**) S_11_; (**b**) VSWR.

**Figure 4 sensors-24-07608-f004:**
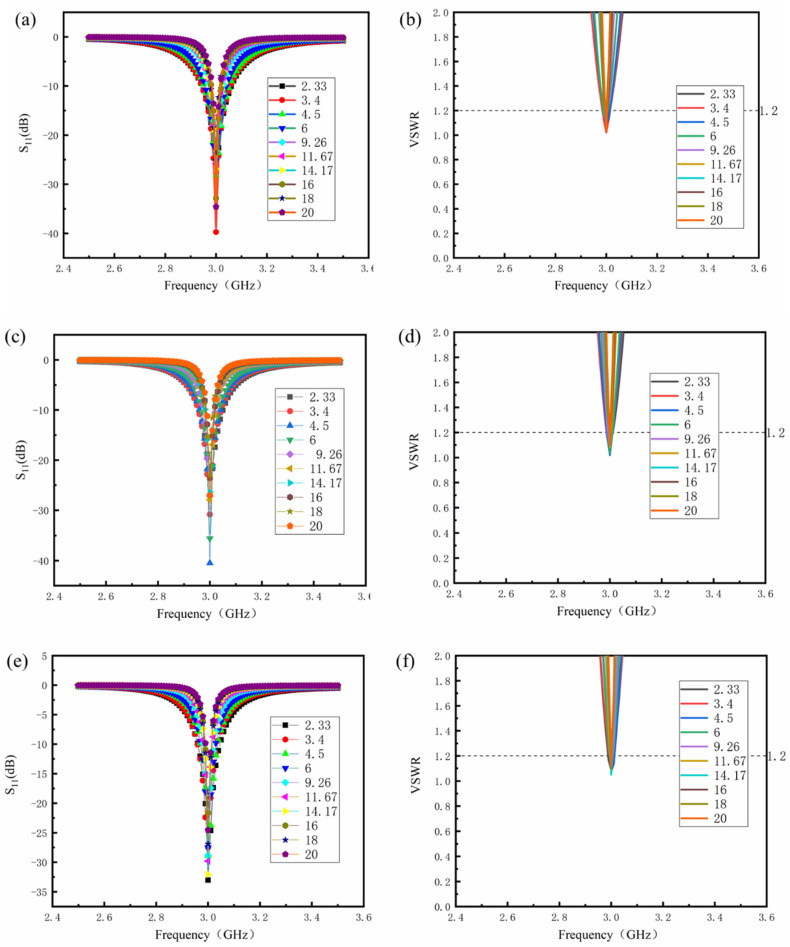
S_11_ and VSWR curves of antenna sensors with varying relative dielectric constant of the dielectric layer at different specified radiation efficiencies: (**a**,**b**) 40%; (**c**,**d**) 50%; (**e**,**f**) 60%.

**Figure 5 sensors-24-07608-f005:**
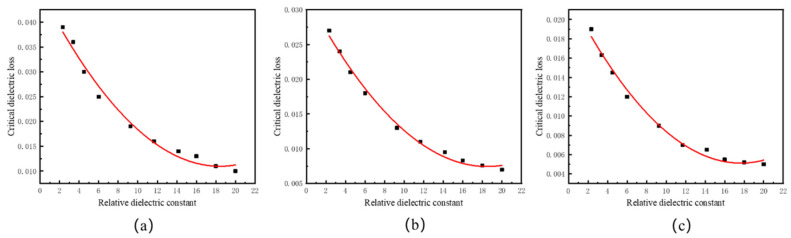
Relationship between relative dielectric constant and critical dielectric loss under predetermined radiation efficiencies: (**a**) 40%; (**b**) 50%; (**c**) 60%.

**Figure 6 sensors-24-07608-f006:**
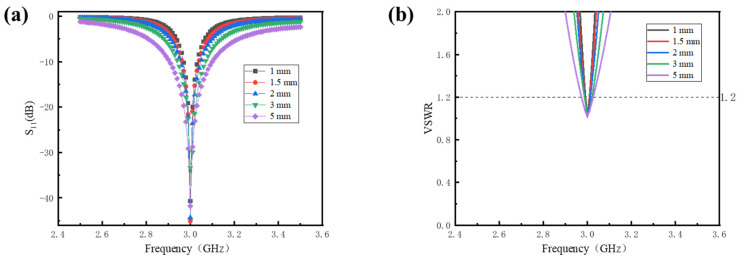
The optimized reflection coefficient of antenna sensors with varying dielectric layer thicknesses: (**a**) S_11_; (**b**) VSWR.

**Figure 7 sensors-24-07608-f007:**
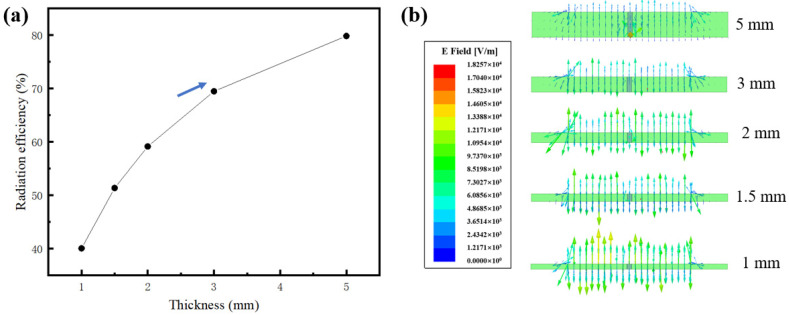
(**a**) Relationship between dielectric layer thickness and antenna radiation efficiency; (**b**) electric field distribution and intensity within dielectric layers of varying thicknesses.

**Figure 8 sensors-24-07608-f008:**
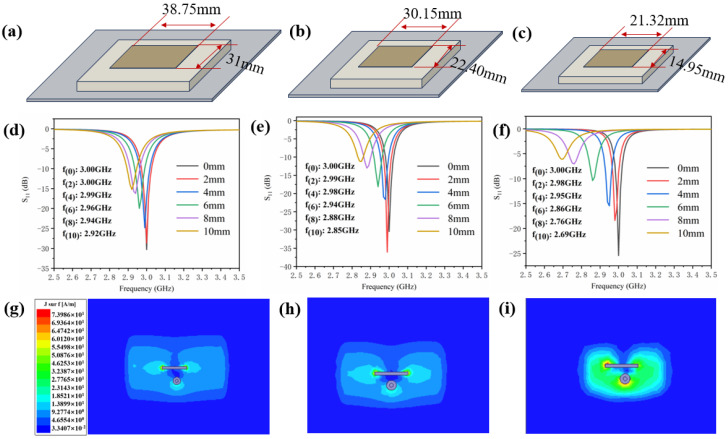
Three different-sized antenna sensors: (**a**) large-sized with Arlon CuClad233 as the dielectric layer; (**b**) medium-sized with Rogers TMM4 as the dielectric layer; and (**c**) small-sized with Arlon AR1000 as the dielectric layer. Relationship between resonant frequency and crack propagation for antenna sensors: (**d**) large-sized; (**e**) medium-sized; and (**f**) small-sized. Surface current density profiles on the ground plane of the antenna sensors: (**g**) large-sized; (**h**) medium-sized; and (**i**) small-sized.

**Table 1 sensors-24-07608-t001:** Relative dielectric constant (ε_r_) and dielectric losses (tanδ) of dielectric layer materials.

Material	Arlon CuClad233	Krempel AKaflex kcl	Rogers TMM4	Rogers TMM6	Arlon AR1000
$\varepsilon_{r}$	2.33	3.40	4.50	6.00	10.00
tanδ	0.0013	0.0018	0.0020	0.0023	0.0030

**Table 2 sensors-24-07608-t002:** Calculated and optimized values of antenna sensor dimensions.

Dielectric Layer Materials	Calculation		Optimization				
	L/mm	W/mm	L/mm	W/mm	L_0_/mm	Antenna Area/mm^2^	Radiation Efficiency/%
Arlon CuClad233	33.16	38.75	31.00	38.75	5.5	1201.25	98.02
Krempel AKaflex kcl	27.25	33.71	25.75	33.71	4.2	868.03	94.97
Rogers TMM4	23.88	30.15	22.40	30.15	3.6	675.36	93.14
Rogers TMM6	21.45	26.73	19.38	26.73	2.8	518.03	90.05
Arlon AR1000	17.11	21.32	14.95	21.32	2.0	318.73	81.57

**Table 3 sensors-24-07608-t003:** Effect of dielectric layer with varying dielectric losses on antenna sensor dimensions and performance.

$\varepsilon_{r}$	tanδ	W/mm	L/mm	$L_{0}/$mm	Radiation Efficiency/%	Peak Gain/dB
4.5 4.5 4.5 4.5 4.5	0.0145 0.0180 0.0210 0.0250 0.0300	30.15 30.15 30.15 30.15 30.15	22.50 22.56 22.59 22.60 22.65	4.6 5.0 5.1 5.5 5.8	59.25 53.77 49.58 45.14 40.65	3.39 3.08 2.84 2.59 2.34

**Table 4 sensors-24-07608-t004:** Antenna sensor dimensions and relationship between the relative dielectric constant and required dielectric loss at the specified radiation efficiencies.

$\varepsilon_{r}$	Radiation Efficiency								
	40%			50%			60%		
	L/mm	L_0_/mm	Required tanδ	L/mm	L_0_/mm	tanδ	L/mm	L_0_/mm	Required tanδ
2.33	31.45	8.30	0.039	31.30	7.20	0.0270	31.20	7.00	0.0190
3.40 4.50 6.00	26.10 22.65 19.60	7.10 6.00 4.80	0.036 0.030 0.025	26.00 22.55 19.50	6.00 5.10 4.00	0.0240 0.0210 0.0180	25.90 22.50 19.50	5.80 4.60 3.80	0.0163 0.0145 0.0120
9.26	15.65	2.90	0.019	15.65	2.70	0.0130	15.60	2.60	0.0090
11.67	13.90	2.42	0.016	13.87	2.36	0.0110	13.85	2.10	0.0070
14.17	12.58	2.10	0.014	12.55	2.00	0.0095	12.52	1.70	0.0065
16.00 18.00 20.00	11.78 11.09 10.48	1.90 1.70 1.50	0.013 0.011 0.010	11.76 11.08 10.45	1.80 1.60 1.40	0.0083 0.0076 0.0070	11.75 11.05 10.45	1.40 1.45 1.20	0.0055 0.0052 0.0050

**Table 5 sensors-24-07608-t005:** Matching of the relative dielectric constant of the dielectric layer with the critical dielectric loss at different specified radiation efficiencies.

$\varepsilon_{r}$	Rated Radiation Efficiency/%					
	40%		50%		60%	
	Required tanδ	R	Required tanδ	R	Required tanδ	R
5	0.0298	39.2	0.0206	47.9	0.0141	58.1
10	0.0183	39.7	0.0127	49.1	0.0084	59.4
15	0.0121	42.3	0.0084	51.6	0.0056	62.1

**Table 6 sensors-24-07608-t006:** Optimized antenna sensor dimensions with different dielectric layer thicknesses.

Layer Thickness /mm	Antenna Sensor Dimensions/mm		
	W	L	L_0_
1	30.43	23.20	5.6
1.5	30.43	22.80	5.0
2	30.43	22.45	4.8
3	30.43	21.80	4.8
5	30.43	21.05	5.8

## Data Availability

The data that support the findings of this study are available from the corresponding author upon reasonable request.

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
