# Peer review of "Effect of Dielectric Layer on Miniaturized Patch Antenna Sensor"

_sensors, 2024, doi:10.3390/s24237608_

Round 1
Reviewer 1 Report
Comments and Suggestions for Authors
Dear Editor,
sensors-3301141,
This study investigates the impact of dielectric layer characteristics on patch antenna sensor miniaturization and performance for crack detection. Higher permittivity (εr) allows for smaller antenna sizes, but increased dielectric losses (tanδ) diminish radiation efficiency. The objective is to identify the optimal combination of εr, tanδ, and layer thickness that minimizes sensor dimensions while maintaining performance. Nonetheless, various aspects necessitate refinement to elevate the manuscript's quality.:
1- The manuscript's title requires refinement for precision.
2- The paper insufficiently explores compromises associated with increasing the dielectric layer's thickness, such as potential size or weight increases that could negate miniaturization benefits.
3- Figures 2, 3, and Table 5 lack adequate captions and explanatory notes.
4- The terms "εr" and "tanδ" are introduced inconsistently, requiring complete forms or definitions upon initial use.
5- The terms in Eq.(1) , Eq.(2), (3), (4) need to be explained for understanding.
6- The abstract contains redundant information and repetitive phrasing, necessitating streamlining for conciseness and readability.
7- The results section would benefit from a comparative analysis of various dielectric materials' performance.
8- The reference format should align with the Sensors journal guidelines.
9- The conclusion reiterates points from the abstract instead of offering an insightful summary, potential applications, or future research directions, particularly regarding the practical use of various dielectric materials in industrial environments.
Author Response
Thank you for your comments. The comments are valuable and helpful for improving the manuscript. Revised portions are marked in red on the paper.
1. The manuscript's title requires refinement for precision.
Response1: Thank you for your valuable comments. After careful consideration, we think this title is suitable for the content of the manuscript.
2. The paper insufficiently explores compromises associated with increasing the dielectric layer's thickness, such as potential size or weight increases that could negate miniaturization benefits.
Response 2: The thickness of the dielectric layer has no obvious effect on the size of the sensor, but the radiation efficiency of the sensor decreases significantly with the decrease of the thickness of the dielectric layer. However, according to the simulation results, the electric field distribution is more concentrated in the thinner dielectric layer. Therefore, under the premise of ensuring sufficient radiation efficiency, the thin dielectric layer should be taken as much as possible to obtain high performance sensors. The analysis and discussion have been supplemented in detail in the paper.
3. Figures 2, 3, and Table 5 lack adequate captions and explanatory notes.
Response 3: Figures 2, 3, and Table 5 captions have been revised. Other figure or table captions have also been checked and revised.
4. The terms "εr" and "tanδ" are introduced inconsistently, requiring complete forms or definitions upon initial use.
Response 4: Thank you for your valuable comments, we have corrected it in the text.
5. The terms in Eq.(1) , Eq.(2), (3), (4) need to be explained for understanding.
Response 5: Thank you, we have revised it in the text to make it more concise for understanding.
6. The abstract contains redundant information and repetitive phrasing, necessitating streamlining for conciseness and readability.
Response 6: The abstract has been revised.
7. The results section would benefit from a comparative analysis of various dielectric materials' performance.
Response 7: Thank you for your valuable comments, we have added it in the text.
8. The reference format should align with the Sensors journal guidelines.
Response 8: The reference format has been revised.
9. The conclusion reiterates points from the abstract instead of offering an insightful summary, potential applications, or future research directions, particularly regarding the practical use of various dielectric materials in industrial environments.
Response 9: Thank you for your valuable comments, we have revised it in the conclusion.
Reviewer 2 Report
Comments and Suggestions for Authors
The paper investigates the influence of dielectric layer properties, specifically permittivity and dielectric loss, on the miniaturization and performance of patch antenna sensors. The authors employ simulations to explore how variations in the dielectric layer affect antenna size, bandwidth, and radiation efficiency. The study provides insights into the impact of dielectric material properties on achieving both compactness and efficiency in patch antenna sensors. While the research is promising, several aspects require further clarification and improvement to strengthen the scientific contribution.
- The authors appear to have mixed up permittivity and relative permittivity of dielectrics. It is unclear which value was used in the calculations, and this distinction could significantly affect the results.
- The antenna area is determined by the operating frequency and dielectric constant according to the equations provided by the authors. Why didn’t the authors consider adjusting the frequency as a method to achieve miniaturization?
- What are the actual materials behind the commercial names of the dielectric materials studied?
- The dielectric constant and dielectric loss of dielectric materials are also frequency-dependent. When specifying values for the dielectric constant and loss, the frequency should be indicated. Additionally, the authors should consider incorporating this frequency dependency into their simulations.
- What are the governing equations or physical processes underlying the relationship between dielectric loss and radiation efficiency? It appears that the authors only provided calculated values without detailed explanations. More details on the underlying mechanisms are needed.
- Experimental validation is essential. Currently, the paper relies solely on mathematical calculations. Including experimental data would greatly enhance the credibility and scientific impact of the findings.
Author Response
Thank you for your comments. The comments are valuable and helpful for improving the manuscript. Revised portions are marked in red on the paper.
1. The authors appear to have mixed up permittivity and relative permittivity of dielectrics. It is unclear which value was used in the calculations, and this distinction could significantly affect the results.
Response 1: Thank you for your valuable comments. We use HFSS software to simulate the effect of dielectric layer on the size and radiation efficiency of the antenna. The dielectric layer materials used are all selected from the software system, and their relative permittivity values are used to calculate the antenna size. The permittivity involved in the article is replaced by the relative dielectric constant, and has been modified.
2. The antenna area is determined by the operating frequency and dielectric constant according to the equations provided by the authors. Why didn’t the authors consider adjusting the frequency as a method to achieve miniaturization?
Response 2: Although increasing the operating frequency can reduce the size of the antenna from the equations, it is not the best way to miniaturize the antenna. The main reason is that the increase in operating frequency may bring more propagation loss, lower efficiency, higher cost, and more complex design issues. In general, operating at larger frequencies, the wavelength of high-frequency signals is shorter, which makes them more susceptible to phenomena such as reflection and refraction during propagation, thus brings propagation loss problems. In addition, the radiation pattern of the high-frequency antenna will become more concentrated, and the radiation energy is mainly concentrated in a small angle range, and the radiation efficiency may also become worse. And it is not possible to increase the frequency infinitely in order to get a smaller size sensor. The influence of the dielectric layer on the sensor should be considered, and the dielectric material should be selected reasonably to meet the design requirements.
3. What are the actual materials behind the commercial names of the dielectric materials studied?
Response 3: The five kinds of dielectric materials are provided by HFSS software. Arlon CuClad233 is a polytetrafluoethylene (PTFE) composite with a copper-plated laminate with a bond layer or prepreg. Krempel AKaflex kcl is a composite of glass fiber and epoxy resin. Both Rogers TMM4 and Rogers TMM6 are ceramic-filled hydrocarbon thermosetting materials based on fluoropolymers. Arlon AR1000 is a polytetrafluorethylene (PTFE) composite reinforced with cross-composite woven glass fiber and micro-dispersed ceramic particles.
4. The dielectric constant and dielectric loss of dielectric materials are also frequency-dependent. When specifying values for the dielectric constant and loss, the frequency should be indicated. Additionally, the authors should consider incorporating this frequency dependency into their simulations.
Response 4: The relative dielectric constant of the layer materials used are all obtained from the software. The measure condition of the relative dielectric constant is generally at the frequency of 1kHz. All the five dielectric materials have their specific relative dielectric constant values. The values of each dielectric material have a different change law with the change of frequency. However, this work is mainly to compare the influence of the values on the sensor size and performance, and does not involve the change law of the dielectric constant and loss with frequency. Therefore, although the operating frequency used in the calculation is set to 3GH, the fixed relative dielectric constant value of each dielectric layer material is used in the calculation and optimization process. Thank you for your suggestion, the frequency corresponding to the relative dielectric constant has been added to the text, and we will further consider the influence of frequency on dielectric constant and loss in the subsequent design and use of devices.
5. What are the governing equations or physical processes underlying the relationship between dielectric loss and radiation efficiency? It appears that the authors only provided calculated values without detailed explanations. More details on the underlying mechanisms are needed.
Response 5: The relationship between the radiation efficiency (η) , relative dielectric constant (εr), and dielectric loss (tanδ) of antenna sensors can be explained by Eq.(6) and (7) in the revised manuscript.From the equations, it can be seen that radiation efficiency is inversely proportional to dielectric loss. Specifically, an increase in dielectric loss usually leads to a decrease in radiation efficiency. By optimizing the relative permittivity and loss factor of the dielectric layer, the radiation efficiency of the antenna can be improved. The relationship between dielectric loss and radiation efficiency has been added in the text.
6. Experimental validation is essential. Currently, the paper relies solely on mathematical calculations. Including experimental data would greatly enhance the credibility and scientific impact of the findings.
Response 6: Thank you for your advice. The main purpose of this work is to guide the selection of suitable dielectric layer materials. In the following work, we will carry out the preparation of dielectric materials and use them for miniaturized antenna sensors.
Reviewer 3 Report
Comments and Suggestions for Authors
The contribution of this paper is very limited. The finding equations are just fitting curves based on the simulation data.
In fact, high permitivity constant results in narrow operating BW and smaller antenna size. Meanwhile, high dielectric loss leads to smaller antenna gain or radiation efficiency.
Author Response
The contribution of this paper is very limited. The finding equations are just fitting curves based on the simulation data. In fact, high permitivity constant results in narrow operating BW and smaller antenna size. Meanwhile, high dielectric loss leads to smaller antenna gain or radiation efficiency.
Response: Thanks for your comments. This work investigates the effect of the dielectric layer on the size and performance of the patch antenna sensor. The high permittivity of the dielectric layer results in a smaller antenna size, but the high dielectric loss also results in a smaller radiation efficiency.​ This study determined the optimal dielectric performance to achieve a balance between size miniaturized and radiation efficiency by analyzing the relationship between permittivity and dielectric loss. This provides guidance for the selection of appropriate dielectric layer materials. We also find that increasing the thickness of the dielectric layer can improve the bandwidth and radiation efficiency. However, as the thickness of the dielectric layer decreases, the electric field distribution in the layer will become more concentrated, which helps to generate stable electromagnetic waves. Therefore, it is advisable to select the thinnest possible dielectric layer while ensuring adequate radiation efficiency. These conclusions provide useful guidance for the selection of effective dielectric layers, which is helpful for the miniaturization design of high-performance antenna sensors.
Round 2
Reviewer 1 Report
Comments and Suggestions for Authors
Dear Authors,
Thank you for your submission. Kindly revise the manuscript by implementing the corrections and removing any unnecessary Word formatting features. Please ensure that all changes are clearly highlighted in yellow for easier review.
Author Response
Thank you for your submission. Kindly revise the manuscript by implementing the corrections and removing any unnecessary Word formatting features. Please ensure that all changes are clearly highlighted in yellow for easier review.
Response1: All unnecessary formatting has been removed, and modifications have been highlighted in yellow. These updates include changes to the abstract, conclusion, captions for Figures 2 and 3, and Table 5, etc. Additionally, the explanations for Eq. (1), Eq. (2), Eq. (3), and Eq. (4) have been revised and supplemented for clarity and comprehensiveness. All reference formats and English grammar have been revised. We sincerely appreciate your valuable comments and are grateful for your dedicated effort and thoughtful feedback.
Reviewer 2 Report
Comments and Suggestions for Authors
The authors still mixed up the concepts of relative permittivity and dielectric constant, which is very unprofessional. The frequency dependent Dk and loss have not been taken into account in the revised manuscript. If there is no experimental validation, more insights and efforts have to be included in the study, such as the suggestions in my first round review.
Author Response
The authors still mixed up the concepts of relative permittivity and dielectric constant, which is very unprofessional. The frequency dependent Dk and loss have not been taken into account in the revised manuscript. If there is no experimental validation, more insights and efforts have to be included in the study, such as the suggestions in my first round review.
Response : The dielectric constant is a property of a material, it indicates the degree of electrical polarization of the material exposed to an electric field and reflects the ability of the material to store electrical energy. The relative permittivity, denoted as εr in the manuscript, is the ratio of a material's dielectric constant to that of a vacuum, and it reflects the material's responsiveness to an electric field. This value is frequency-dependent, as the material's ability to respond to an electric field varies with changes in molecular or ionic movement. At high frequencies, εr fluctuates due to altered molecular or ionic response capabilities, while at low frequencies, it remains relatively stable. The working frequency used in the manuscript is less than 3 GHz, and the relative permittivity of the five dielectric layer materials used in the manuscript is in a stable range, which can be regarded as a stable value. In fact, the relative permittivity is affected by temperature, pressure, humidity and other factors in addition to frequency, which makes the change of εr under these factors complicated. However, there is still a lack of a clear model that can take these factors into account, and the relative permittivity of various materials with the change of external factors is also significantly different. Therefore, the size calculation equations for the antenna sensor provided in the literature [19], [20] are based on a stable value of the relative permittivity. Building on this, further size and performance optimizations are carried out using HFSS software. The purpose of the simulation work is to identify suitable dielectric layer materials for compact patch antenna sensors with high radiation efficiency, while also providing targets and guidance for the subsequent development of advanced materials for crack propagation. Additionally, we analyzed and compared the effects of different antenna sensor sizes on crack detection to enhance the completeness of the manuscript, as highlighted in yellow in section 3.5. We sincerely appreciate your valuable comments and are grateful for your dedicated effort and thoughtful feedback.
Reviewer 3 Report
Comments and Suggestions for Authors
The paper can be accepted in current form.
Author Response
The paper can be accepted in current form.
Response: We are grateful for your dedicated effort and thoughtful feedback.